# EBIND: A PRACTICAL APPROACH TO SPACE BINDING

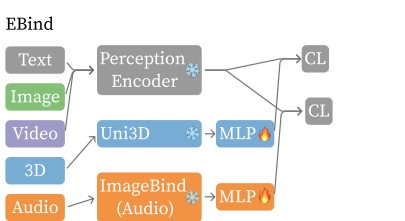

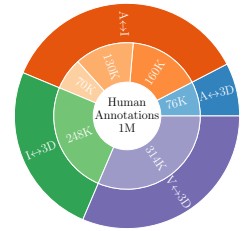

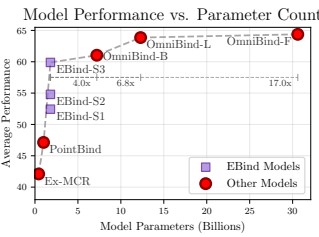

(a) Model Overview.
CL means Contrastive Loss.

(b) New Human Annotations.

(c) Avg. over 13 benchmarks.

Figure 1: Overview of the `EBind` approach and results.

## ABSTRACT

We simplify space binding by focusing on two core components, a single encoder per modality and high-quality data; enabling training state-of-the-art models on a single GPU in a few hours as opposed to multiple days. We present `EBind`, an Easy, data-centric, and parameter-efficient method to `Bind` the embedding spaces of multiple contrastive models. We demonstrate that a simple 1.8B-parameter image-text-video-audio-3D model can outperform models 4 to 17× the size. The key to achieving this is a carefully curated dataset of three complementary data sources: i) 6.7M fully-automated multimodal quintuples sourced via SOTA retrieval models, ii) 1M diverse, semi-automated triples annotated by humans as negative, partial, or positive matches, and iii) 3.4M pre-existing captioned data items. We use 13 different evaluations to demonstrate the value of each data source. Due to limitations with existing benchmarks, we further introduce the first high-quality, consensus-annotated zero-shot classification benchmark between audio and PCs. In contrast to related work, we will open-source our code, model weights, *and* the datasets.

## 1 INTRODUCTION

Multimodal contrastive models have emerged as foundational components of modern AI systems. From CLIP's revolutionary image-text alignment (Radford et al., 2021) to CLAP's audio-text embeddings (Elizalde et al., 2023) and Uni3D's 3D-text representations (Zhou et al., 2024), these models power advanced machine learning applications like retrieval systems (Abootorabi et al., 2025) and automatic labeling (Gao et al., 2024; Zhang et al., 2024a) to conditional generation (Li et al., 2023; Ramesh et al., 2022; Steiner et al., 2024; Guo et al., 2023). Due to the enablement of processing many modalities, like text, images, audio, video, and 3D point clouds (PC), multimodal learning is increasingly recognized as a critical enabler of progress toward artificial general intelligence (Song et al., 2025).

The natural progression toward truly unified multimodal understanding has led researchers to explore binding these separate bi- or trimodal embedding spaces into joint representation spaces, as demonstrated by pioneering work such as ImageBind (Girdhar et al., 2023), LanguageBind (Zhu et al., 2024), and OmniBind (Kong et al., 2025). Binding multiple modalities together demonstrates emergent properties like the ability to do similarity searches between any two modalities.

In this work, we pay extra attention to models that embed not only vision and language but also audio and PCs, e.g., Zhang et al. (2024b) and Wang et al. (2025). Such multimodal models not only have the potential to power next-generation LLMs, retrieval systems, and generative models spanning more modalities, but also to help advance fields like autonomous driving (Shao et al., 2022), 3D scene understanding (Vu et al., 2022), and robotics (Huang et al., 2023). However, despite their theoretical appeal and demonstrated feasibility, the field faces several limitations that hinder widespread adoption and rapid development.

**Current Limitations** are associated with data, compute, and evaluations. First, there is a lack of easy-access *data* for doing contrastive learning across the five modalities audio, image, video, text, and PC in combination. The scarcity causes research to focus on methodologies that treat modalities separately (Guo et al., 2023) or build artificially paired data via retrieval models (Zhang et al., 2024b; Wang et al., 2025). The latter methodology is critical as, to the best of our knowledge, there exist no such paired datasets in the open-source community. In turn, progress slows down and reproducibility becomes impossible.

Second, from a *computational* perspective, current approaches either suffer from excessive resource requirements or lack performance. Some models range from 7-30B parameters (Kong et al., 2025; Wang et al., 2025), creating a barriers to entry for research groups with fewer resources and limits the democratization of multimodal AI research. Other smaller sub-1B parameter models sometimes perform $3-4\times$ worse (Guo et al., 2023; Wang et al., 2023; Zhang et al., 2024b).

Finally, the *evaluation* landscape presents additional challenges. Particularly pronounced are issues with PCs. Benchmarks predominantly remain limited to synthetic benchmarks rather than real-world scenarios (Deitke et al., 2023; Wu et al., 2015) and, audio-PC does not yet have a benchmark. Furthermore, multiple publications have argued that testing on data that the underlying model was trained on is okay (Wang et al., 2025; 2021) which, to us and many others, is wrong.

**Our Contributions: EBind.** In response to these challenges, we present `EBind`, a simple yet effective model and methodology that achieves state-of-the-art (SOTA) performance for its compact 1.8B-parameter size, often outperforming models 4-17 times larger. Our key insight centers on prioritizing two core components: i) employing a simple, well-chosen model architecture and ii) leveraging carefully curated, high-quality training data.

`EBind` democratizes multimodal model training by making it possible to obtain SOTA results on a single GPU within hours rather than days of distributed training. This efficiency stems from a simple choice of model and training scheme that has a low memory footprint, coupled with a data-centric approach, which we validate through comprehensive empirical analysis.

We introduce a systematic three-tier data curation strategy. Inspired by a mix of related work (Zhang et al., 2024b; Wang et al., 2025; Kong et al., 2025; Girdhar et al., 2023), we construct our training corpus from: (1) 6.7M fully-automated multimodal quintuples generated using state-of-the-art retrieval models, (2) 3.4M high-quality pre-existing captioned data items, and (3) 1M human-annotated samples with explicit positive, negative, and partial match labels. This principled approach to data construction enables us to systematically study the contribution of each data source to the final model performance.

To address evaluation limitations, we further develop a first-of-its-kind, high-quality, consensus-annotated evaluation benchmark that combines PCs and audio in a zero-shot task. Unlike previous work, we commit to full transparency by open-sourcing our complete codebase, trained model weights, *and* curated datasets, enabling true reproducibility and fostering further research.

Through comprehensive evaluations across 14 datasets spanning all five modalities, we demonstrate the individual value of each component of our data curation strategy. Our experiments validate the central hypothesis that data quality and careful curation strategies can achieve superior performance compared to architectural complexity and scale. We show that our simpler approach can often match or exceed the performance of much larger, more complex models. This holds across diverse multimodal understanding tasks.

## 2 Related Work

Contrastive representation learning, pioneered by the CLIP model (Radford et al., 2021), establishes a shared embedding space between modalities. While the shared embedding space of CLIP, and later advancements like SigLIP (Zhai et al., 2023) and Perception Encoder (Bolya et al., 2025), cover vision and text, a natural expansion of the field has been to adapt the contrastive learning approach to, match text and audio (Elizalde et al., 2023; Mei et al., 2024) or text and PCs (Xue et al., 2023; Zhou et al., 2024).

Successively, multiple such bi-modal models have been joined into compositions that unify the individual modalities. These efforts, often referred to as "space binding methods," face the challenges of acquiring sufficient paired data and managing architectural complexity. In contrast to recent trends, which prioritize model scale and intricate training procedures, our approach, `EBind`, aims to keep model size and complexity low.

### 2.1 Model Composition and Training Algorithms

Recent research in achieving unified multimodal representations has trended toward integrating multiple specialized (and often frozen) encoders. Here, we split approaches into two. On one side, most methodologies are based on a composition of one encoder per modality (Wang et al., 2023; Zhang et al., 2024b; Girdhar et al., 2023; Zhu et al., 2024; Guo et al., 2023). These are all sub-1B parameter models and perform significantly worse than the other approaches. On the other side are the OmniBind models from (Wang et al., 2025). The authors present three model compositions ranging from 7 to 30B parameters where more than one model per modality is employed. Among models that can embed all text, image, audio, video, and PCs, the largest of the OmniBind models constitutes the SOTA. In this work, we demonstrate that it is possible to get most of the performance from the bigger models with one frozen encoder and an MLP projector per modality and $4\times$ fewer parameters.

When considering training complexity, ImageBind and LanguageBind are amongst the simpler (Girdhar et al., 2023; Zhu et al., 2024). They train full, unfrozen models with the InfoNCE loss (van den Oord et al., 2018) against their base modality (image and text, respectively) (Girdhar et al., 2023; Zhu et al., 2024). In the other end of the spectra, the large-scale OmniBind models require intricate mechanisms to combine multiple embedding spaces effectively (Wang et al., 2025). The models employ a routing strategy (inspired by Mixture-of-Experts (Mu & Lin, 2025)) to dynamically weight contributions from different models. They require complex objectives, including a cross-modal overall alignment loss and a language representation decoupling loss to mitigate conflicts between different text embeddings. C-MCR Wang et al. (2023) combines Gaussian noise injection with an InfoNCE contrastive loss (van den Oord et al., 2018) complemented by a semantic-enhanced inter- and intra-model connection method. Ex-MCR Zhang et al. (2024b) extends upon that and utilizes a dense InfoNCE loss across all modality pairs, alongside an additional $L_2$ loss.

Albeit not a binding method in exactly the same spirit as the above, we further found some inspiration from the UNITE method Kong et al. (2025). The method introduces the a variant of the InfoNCE loss where separate modalities are separated in the loss computations. Similarly, we notice that splitting entire batches into isolated modality pairs, and even isolated tasks, works well. In contrast to most related work, our method requires neither complex masking schemes or loss compositions.

### 2.2 Data for Multimodal Alignment

While the underlying encoders of the "binding approaches," e.g., CLIP, CLAP, and Uni3D (Zhou et al., 2024), require large-scale datasets, typically sourced from the Internet, we focus on data requirements for binding models. Most multimodal binding models match a few modalities and observe models associate unmatched modalities as an emergent property. ImageBind (Girdhar et al., 2023) learns a joint embedding across six modalities (Image, Text, Audio, Depth, Thermal, and IMU) by individually pairing each to images. LanguageBind (Zhu et al., 2024) and UNITE (Kong et al., 2025) both use language as the

semantic anchor to bind modalities. Other methods like EX-MCR and OmniBind use various forms of retrieval to synthetically compose "pseudo pairs" (Zhang et al., 2024b; Wang et al., 2025). We further notice that most high-performing models employ video in their training datasets which is, surprisingly, not the case for the OmniBind models.

We combine learnings from related work by leveraging both pseudo pairs and video data. `EBind` achieves SOTA results by relying on a systematic three-tier data curation strategy: 1) fully-automated multimodal "pseudo quintuples" (5-tuples) generated by SOTA retrieval models, 2) semi-automated, human-verified data, and 3) pre-existing open-source captioned data. Our focus on high-quality, curated training data and a simple model composition allows us to maintain a simple training scheme where batches only require the projected modality and one or more of the frozen modalities.

## 3 Open Dataset

### 3.1 Training Datasets

We now give a high-level description of our procedure to build the dataset. Further details are deferred to Appendix A. Our full dataset is a composition of three splits. Section 3.1.1 describes the fully-automated first split, which attempts to follow the approach from Wang et al. (2025) to construct 6.7M quintuples of all five modalities (audio, image, PCs, videos, and captions). Section 3.1.2 details the second split; a semi-automated, human-verified split of 1M triples spanning captions and two other modalities. Section 3.1.3 describes our third split; a collection of open-source datasets comprising 3.4M already captioned data items. It should be noted that the underlying data from the three splits overlap.

### 3.1.1 Split 1: Automatically Paired 5-tuples (6.7M)

For the first phase of training, we follow Wang et al. (2025) to build an automatic retrieval-based dataset. We do this by sourcing captions from 12 different datasets, as detailed in Table 6. We ignore the pairings between the various modalities of our source data and treat each as a separate unimodal dataset. After deduplication, we source 6.7M text captions. We similarly merge audio, image, video, and PC data, from the datasets listed in Table 7. We then pair each text caption with the best matches in the other four modalities via SOTA bi-modal embedding models listed in Table 10. In turn, we retrieve the nearest neighbor for each modality, and construct 6.7M 5-tuples of the form (text, image, video, audio, PC) for each text caption.[1] We employ graph-based `HNSW32` indices from FAISS Douze et al. (2024) to perform retrieval. The number of unique items sourced from each underlying dataset to build Split 1 is detailed in Table 8.

To elevate the quality of Split 1, we go through multiple filtering steps to avoid train-val leakage as well as inaccessible or duplicate data. We ensure that all captions sourced from datasets with a test-train split are sourced from their train sets only. Furthermore, we ensure that no occurences of VGG-SS (Chen et al., 2021), Audioset eval (Gemmeke et al., 2017), AudioCaps (Kim et al., 2019), or Objaverse-LVIS (Deitke et al., 2023) points appear in our retrieval databases. [2]

### 3.1.2 Split 2: Human verified triples (1M)

While examining captions from Split 1, we notice that some modalities have more natural captions than other. For example, looking at PC captions from OpenShape (Liu et al., 2023), we find examples like "a fish is shown on a black background" and "a 3d model of a rock with a hole in it." Arguably, such captions are less relevant for real-world use-cases. As a consequence, we devise a methodology for efficiently collecting more human-verified pairs.

---

[1] A tempting idea is to retrieve $k > 1$ neighbors per caption to increase the dataset size. We tried that but found no particular benefit from it.

[2] In the reviews of the OmniBind paper (https://openreview.net/forum?id=l2izo0z7gu) concerns were raised that, e.g., AudioCaps may cause leakage issues.

We run several human annotation projects to verify captions from one dataset are appropriately automatically paired with data items from other datasets. We start by selecting a subset of captions of the datasets listed in *italic* in Table 4 in the appendix and retrieve up to eight neighbors of the modality to be added from the other dataset. We filter the matches to ensure diversity, selecting three candidates per text caption. We ask human annotators to label each candidate as a positive, partial, or negative match to the text caption shown (for details, see Appendix A.3.) We retain all annotations, including partial and negative matches, and use them in training as described in Section 4. We find the partial and negative annotations to be helpful as hard negatives. The annotators label a total of 332K captions (1M annotations). The exact numbers are reported in Table 5 in the Appendix. Figure 1b on the front page displays the data composition. The details of pre-processing, data pairing, and human verification can be found in Appendix A.

### 3.1.3 Split 3: Open-source captioned datasets (3.4M)

For some modality pairs, we identify that there is a benefit to using data with original captions. The two most pronounced use-cases are video datasets that have a natural correspondence to audio and PCs that come naturally with their 3D renders as images. We source the data listed in Table 9 and use the modalities listed. For example, the table shows that we use the PCs from OpenShape (Liu et al., 2023) together with their renders and we use both the vision and the audio part from videos datasets like VGGSound (Chen et al., 2020) against the same caption. We are careful to remove any leakage to evaluation sets.

### 3.2 Zero-Shot PC–Audio Evaluation Dataset

At present, the field of multimodal retrieval models is lacking in benchmarks for new modality pairs. Even for the relatively popular point–audio modality pair, where models such as OmniBind and Ex-MCR can perform cross-modal retrieval, no evaluation dataset currently exists. In addition, most public evaluation sets for retrieval between PCs and other modalities are synthetic, limited in scope, and hardly reflective of real-world performance.

We make progress on this front by creating a realistic zero-shot points–audio evaluation dataset `EShot`. This is the first of its kind, to the best of our knowledge.

We sample both modalities from evaluation splits of public datasets; audio from AudioSet, and PCs from Objaverse-LVIS. As with the creation of our training sets, we automatically pair the two modalities through text captions, using SOTA retrieval models. We then run every pair, through a human consensus check forcing three individual annotators to agree on each pair. This results in 1775 and 1763 unique audio and PC items, respectively, which we use for zero-shot classification. We do so by deriving 112 classes from the PC's Objaverse-LVIS categories (this is detailed in Appendix A), and define a zero-shot classification task as follows: We group the audio items by class, and take the mean embedding of each class as the class representative. Each PC is then classified based on its embedding's similarity to that of the class representative. Swapping the roles of the modalities results in a classification task for audio. Note that this classification task is similar to the common practice of using multiple text prompt templates for zero-shot modality-to-text classification tasks.

## 4 Model and Training

Our model is simple; see Figure 1a. It consists of a frozen, pre-trained encoder for each modality and projectors for just audio and PCs. Keeping the encoders frozen keeps the computational complexity and memory requirements low, since we can extract and store embeddings ahead of training. Freezing the vision and text encoders is particularly advantageous, as it enables the incremental addition of other modalities to the model, independently of one another. The text (353M parameters) and vision (317M) encoders in our model are from the Perception Encoder's $PE_{core}L$ variant (Bolya et al., 2025). The vision encoder handles both images and videos, uniformly sampling 8 images from videos and averaging their output embeddings. The audio encoder (90M) is from ImageBind (Girdhar et al., 2023), and the PC encoder (1.02B) is the Uni3D Zhou et al. (2024) variant trained without

the Objaverse-LVIS dataset. We restrict the training to projectors at the output of the audio and points encoders. Both projectors are two-layer multilayer perceptrons with 1024 dimensions at the input and the output, and 2048 dimensions at the hidden layer, adding 4.2M parameters each and bringing the total parameter count to 1.79B.

In order to simplify the addition of new modalities to the model, we train each projector separately and make no effort to further align the trained projectors with each other. We only require each batch of data to contain the projected modality and one or more of the frozen modalities, which adds flexibility and simplifies sourcing more training data. As a consequence, we take less than full advantage of our retrieval-based 5-tuple training data, as only one of the two projected modalities enters training each projector.

We train the model contrastively, using a loss function that exploits the positive, partial, negative match labels in our human-annotated data. We do so by assigning target probability $p = 1$ to positive matches, $p = 0.5$ to partial matches, and $p = 0$ to negative matches, and minimizing the cross entropy loss between the target probability and the model's prediction. That is, given a batch of output embeddings $a_1^B$ for a projected modality and the paired output embeddings $b_1^B$ for a frozen modality, we define the loss as

$$L(a_1^B, b_1^B) = -\frac{1}{B} \sum_{i=1}^{B} p_i \log q_i + (1 - p_i) \log(1 - q_i)$$

where the predicted probability $q_i$ for the $i$th pair is defined in the usual way, as

$$q_i = \frac{e^{a_i^T b_i / \tau}}{\sum_{j=1}^{B} e^{a_i^T b_j / \tau}}$$

Here $\tau$ is a trainable and modality-pair-dependent temperature parameter. The total loss for the batch is then defined as

$$L(\text{batch}) = \sum_{j=1}^{m} L(a_1^B, b_{j1}^B) + L(b_{j1}^B, a_1^B)$$

where $m$ denotes the number of frozen modalities in the batch, and $b_{j1}^B$ denotes the embeddings of the $j$th frozen modality.

We stage the training data in order of increasing pairing quality and decreasing model-richness. That is, we first train on Split 1, our 6.7M automatically-paired 5-tuples; followed by Split 2, the 1M human-verified triples; and finally on Split 3, the 3.4M captioned triples. Each stage consist of 2 epochs of training. `EBind`'s structure allows us to extract and store the output embeddings from the underlying encoders and only load the embeddings and the projectors at training time. This, in turn, allows us to fit a batch size of 2048 on a single A100 GPU with 40GB of memory, and train to complete in less than 4 hours. We use the AdamW optimizer with an initial learning rate of 0.001 and cosine annealing scheduler. We initialize all temperatures to $\tau = 0.07$.

## 5 EVALUATION

Similar to related work, we evaluate `EBind` on 13 different public benchmarks (See Table 11 in Appendix C for an enumeration). We report scores for three versions or our model; `EBind`-S1, trained on our Split 1; `EBind`-S2, trained on Split 1 and 2; and `EBind`-S3, trained on all three splits. On single NVIDIA A100 GPU with a 30-core 216GB RAM CPU, it takes 15, 30, and 15 minutes, respectively, to train a model for epochs on each split. Further details on non-trivial evaluations are provided in Appendix C. We compare `EBind` against other text-image-audio-PC models; EX-MCR (Zhang et al., 2024b), PointBind (Guo et al., 2023), and OmniBind (Wang et al., 2025). We note that as Omnibind-L and OmniBind-F use a variant of Uni3D that was trained on Objaverse-LVIS (Deitke et al., 2023), those may contain train-test leakage. We further report the performance of `EBind` on `EShot` to establish

Table 1: Cross-modal retrieval results ordered by model size. Model performance for models other than `EBind` are sourced from Wang et al. (2025). Best and second best result in **bold** and underlined, respectively. * may contain train-test data leakage.

| Models | Size (B) | Audio-Text | | | | Audio-Image | | | | Image-Text | | | | Points-Image | |
| | | AudioCaps | | Clotho | | VGG-SS | | FlickrNet | | COCO | | Flickr30K | | Objaverse | |
| | | R@1 | R@5 | R@1 | R@5 | R@1 | R@5 | R@1 | R@5 | R@1 | R@5 | R@1 | R@5 | R@1 | R@5 |
|---|---|---|---|---|---|---|---|---|---|---|---|---|---|---|---|
| Ex-MCR | 0.43 | 19.07 | 47.05 | 7.01 | 22.04 | 2.13 | 8.13 | 1.57 | 5.94 | 40.24 | 64.78 | 71.89 | 90.55 | 2.54 | 8.25 |
| PointBind | 1 | 9.24 | 27.47 | 6.64 | 17.28 | 14.82 | 35.67 | 7.68 | 20.79 | 57.28 | 79.54 | 86.04 | 96.97 | 5.86 | 14.59 |
| EBind-S1 | | 13.38 | 39.60 | 9.35 | 24.65 | 3.97 | 14.50 | 2.53 | 9.21 | | | | | 17.63 | 37.57 |
| EBind-S2 | 1.8 | 17.14 | 45.25 | 7.91 | 22.05 | 11.04 | 30.10 | 5.10 | 16.34 | **65.05** | **84.91** | **90.72** | **98.4** | 16.64 | 36.03 |
| EBind-S3 | | 23.35 | 53.81 | 11.31 | 28.54 | **25.92** | **54.37** | **9.14** | **23.97** | | | | | 46.34 | **70.17** |
| OmniB-B | 7.2 | 43.61 | 76.02 | 20.94 | 46.77 | 14.11 | 35.74 | 7.67 | 21.65 | 56.94 | 80.11 | 85.99 | 97.02 | 34.34* | 58.40* |
| OmniB-L | 12.3 | **47.89** | **79.75** | 23.07 | **49.67** | 14.14 | 36.07 | 7.86 | 21.72 | 60.08 | 82.35 | 87.20 | 97.40 | 46.09* | 69.11* |
| OmniB-F | 30.6 | 46.72 | 79.69 | **23.27** | 49.46 | 15.64 | 38.19 | 8.32 | 23.49 | 62.64 | 83.79 | 89.13 | 97.82 | **46.55*** | 69.92* |

a baseline for future work. To avoid reporting more results with leakage, we refrain from evaluating other models on `EShot`. Since `EBind` inherits its Text-Image performance from Perception Encoder Bolya et al. (2025) we report those only once as they do not change. We also avoid reporting video benchmarks as they will be identical to those in Bolya et al. (2025) and are not comparable to other models in this section. In Section 5.1 and 5.2 below, we focus on our best performing model `EBind`-S3 and keep the analysis of our dataset splits to Section 5.3.

## 5.1 RETRIEVAL

Table 1 shows our performance on zero-shot retrieval tasks. Perhaps not surprisingly, we find that `EBind`-S3 is consistently outperforming models of smaller size (rows above). When comparing the model to those that are larger, we find that except for Audio-Text retrieval tasks, `EBind`-S3 is consistently amongst the two best models. On Image-Text and Audio-Text tasks, it even outperforms a model 17× larger. While the Image-Text results are not surprising, as it is merely "reproductions" of results reported in Bolya et al. (2025), Audio-Image and Points-Image demonstrate how having a strong, frozen backbone can suffice. A particularly notable observation is that Uni3D quotes 45.8 on Objaverse-LVIS for their model *not* trained on the Objaverse-LVIS data and 55.3 for the one *with* that subset. We use the encoder without leakage and surpass the reported performance, indicating that our methodology is not fundamentally upper-bounded by the underlying encoders. Notably, all of the OmniBind model's use a version of Uni3D that *has leakage* and fail to match the underlying performance.

While we have no concrete evidence as to why `EBind` cannot compete with the larger models on Text-Audio retrieval tasks, we do have two compelling hypotheses and an observation that may justify it. We chose to use the audio encoder from ImageBind Girdhar et al. (2023). It was not originally trained with a contrastive loss against text and was optimized against image rather. In turn, the embedding space may not be as easy to project onto that of Perception Encoder. This hypothesis is backed by Girdhar et al. (2023) reporting R@1 at 9.3 on AudioCaps and 6.0 on Clotho. The audio encoder further has less parameters than, the vision encoder (90M vs. 353M). We also observe that while all other modalities are supported by one encoder in OmniBind-B (the smallest version), audio has three underlying encoders, perhaps because one model alone did not work well. Finally, PointBind uses the same audio encoder as us but has lower scores, indicating that our data and training approach may still be relevant for Audio-Text.

## 5.2 ZERO-SHOT CLASSIFICATION

Table 2 displays out zero-shot retrieval results. `EBind` outperforms models of smaller sizes across all benchmarks. When looking at the OmniBind-B model, which is 4× larger than ours, `EBind` performs best in a few cases but generally scores a bit lower. Two points worth noting are that i) six out of the 11 reported numbers are on PC benchmarks where we believe there may be train-test leakage for the OmniBind models and ii) on ImageNet, Bolya et al. (2025) report numbers that exceed those of OmniBind which we could not reproduce.

Table 2: Zero-shot classification results ordered by model size. Model performance for models other than `EBind` are sourced from Wang et al. (2025). Best and second best result in **bold** and underlined, respectively. * may contain train-test data leakage.

| Model | Size (B) | Audio | | | Image | | Points | | | | | |
| | | AudioSet mAP | ESC-50 Top1 | Top5 | ImageNet Top1 | Top5 | Objaverse Top1 | Top5 | ScanObjectNN Top1 | Top5 | ModelNet40 Top1 | Top5 |
|---|---|---|---|---|---|---|---|---|---|---|---|---|
| Ex-MCR | 0.43 | 6.67 | 71.20 | 96.80 | 60.79 | 86.98 | 17.94 | 43.37 | 40.31 | 77.20 | 66.53 | 93.60 |
| PointBind | 1 | 13.96 | 67.25 | 87.50 | 76.13 | 94.22 | 13.83 | 30.34 | 55.05 | 86.89 | 76.18 | 97.04 |
| `EBind`-S1 | | 19.6 | 76.40 | 95.50 | | | 43.40 | 71.91 | 57.75 | 88.48 | 86.46 | 97.38 |
| `EBind`-S2 | 1.8 | 14.34 | 75.80 | 92.40 | 79.16 | 94.92 | 42.08 | 70.95 | 56.92 | 87.27 | 86.12 | 96.80 |
| `EBind`-S3 | | 21.28 | 78.45 | 96.00 | | | 46.56 | 75.52 | 57.65 | 89.20 | 86.56 | 97.48 |
| OBind-B | 7.2 | 21.19 | 92.90 | 99.75 | 76.18 | 94.02 | **53.30*** | 81.85* | 57.79 | 89.76 | 82.82 | 97.12 |
| OBind-L | 12.3 | **25.57** | 93.25 | 99.80 | 78.87 | 95.32 | 53.97* | **82.90*** | **64.67*** | 94.15* | 86.55* | 99.03* |
| OBind-F | 30.6 | 25.14 | **93.45** | **99.85** | **79.93** | **95.86** | 53.56* | 81.82* | **64.67*** | **94.36*** | **87.12*** | **99.03*** |

By inspecting the individual modalities, similar observations can be made to the previous section. Namely, classifying audio is the weakest point, potentially for the same reason as above, and the inherited image classification capabilities are strong. One difference lies in the reported scores from Uni3D on the Point benchmarks. On Objaverse-LVIS and ModelNet40, `EBind`-S3 achieves similar scores to Uni3D (reported R@1 47.2 and 86.6, respectively). For ScanObjectNN, however, `EBind`-S3 scores 57.7 while Uni3D scores 66.5. We have no clear answer as to why this is the case.

Table 3: Zero-shot classification results on `EShot`.

| Models | R@1 | R@5 |
|---|---|---|
| `EBind`-S1 | 56.60 | 79.88 |
| `EBind`-S2 | 64.27 | 85.79 |
| `EBind`-S3 | 57.26 | 86.22 |

**EShot**   To understand the abilities of `EBind` on Audio-Points tasks, we use `EShot`. We report our numbers in Table 3 in the hope that the field will find it useful as to further the understanding of performance of multimodal retrieval models. Below, we further use the benchmark to analyze our dataset splits.

## 5.3 Analysis of Dataset Splits

In this section, we use Figure 1c on the front page, Table 1, and Table 2 to further analyze the effect of the three dataset splits we employ. In Figure 1c, we have averaged all scores from the two tables (not including `EShot`) and plotted them against model sizes. Orthogonal to the findings from OmniBind showing that scaling model parameters leads to better performance, we show that carefully curating the right data can similarly improve performance. Furthermore, the plot indicates that following that fully automated approach of matching data with retrieval models cannot stand alone. Both employing humans to segregate good from bad pairs (Split 2) and using already captioned / naturally paired data (Split 3) improves performance.

From considering the Audio related columns in Table 1 and 2, it is apparent that including Split 2 improves many benchmark, arguably by removing noise from the automatic pairs. However, the Point related benchmarks do generally not see the same effects. We attribute the reason to most existing point-evaluation benchmarks being tied to synthetic renders of PCs, either explicitly or implicitly via the way the benchmarks are constructed. Neither Split 1 not Split 2 contain any PC renders, likely answering why performance remains low.

As indicated by Table 9 in the Appendix, we add captioned 3D PCs and their renders and see improvements on point-related benchmarks, especially on point-image retrieval. Similarly, we add both audio and visuals from many captioned video datasets which further increase both audio-image and audio-text performance. Based on these findings, we hypothesize that adding even more such "naturally paired" data further improve performance.

Finally, we observe that the R@1 performance on `EShot` is negatively affected by Split 3. We attribute this to the fact that Split 1 and 2 have data that pairs audio and PCs while Split 3 does not. It remains an open question how to avoid such a problem of "forgetting."

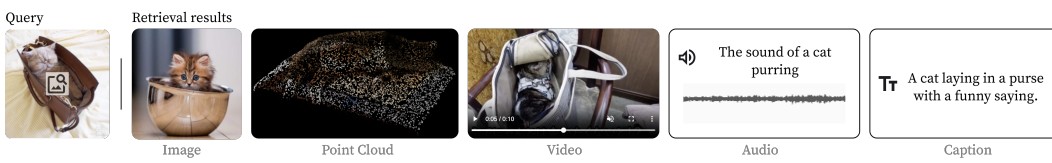

Query | Retrieval results

Image | Point Cloud | Video | Audio | Caption

Figure 2: A top-1 retrieval example from `EBind`-P3.

## 6 APPLICATIONS

Models like `EBind` have multiple applications. In this section, we briefly demonstrate two of them. In Figure 2, we show an example of querying a database of images, point clouds, videos, audio, and captions based on an image. `EBind`-S3 is used to embed the query image of the cat to the left and an entire dataset of the five modalities: text, image, video, audio, and PC. Retrieval is done by the most similar item from the database based on the cosine similarity between the query and every embedding. As shown, the model identifies items from each modality that semantically relate to the query image.

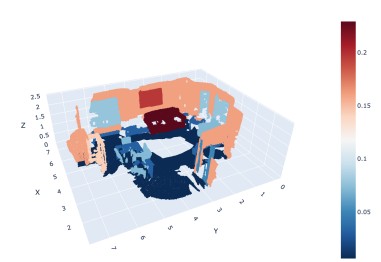

Figure 3: Zero-shot PC classification post segmentation.

In Figure 3, we used `EBind`-S3 to embed pre-segmented objects from a point cloud scene and computed similarities to the word "Sofa." As indicated by the colors (red indicating more similar), we can use the model to identify the object that is most likely to be a sofa. Arguably, such applications could find use in the physical artificial intelligence space.

These examples show how retrieval models, like ours, can have many applications. For more examples, please see Guo et al. (2023) that, builds a multimodal large language model on top of their retrieval model to enable it to "see PCs" and Girdhar et al. (2023) that demonstrates arithmetics on embeddings to fuse an embedding of a traffic sign image with a sound clip of rain into a query to identify images from city streets in rainy weather.

## 7 CONCLUSION

We present `EBind`; an Easy, data-centric, and parameter efficient model that `Bind`s five different modalities into one coherent embedding space. Our core contributions includes achieving SOTA results while avoiding model and training complexity, and a carefully curated dataset that includes both fully automated; semi-automated, human-verified; and pre-existing captioned data. Finally, we introduce a new high-quality, consensus-verified zero-shot classification benchmark `EShot` to help guide future developments within the field.

## 8 FUTURE WORK

We see many potential improvements and applications of this work. Here, we name but a few. First, the human-verified data that we introduce may have even better use. For example, we do not propagate information about dataset statistics or similarity thresholds from Split 2 backward into the data that we assemble with Split 1 to elevate quality. Second, as we see the field of physical artificial intelligence and world-modeling gaining momentum, having truly multimodal models that can understand many sensors becomes increasingly valuable. As a consequence, our work shows that identifying more naturally paired modalities, similar to that of vision and audio in videos and PCs scanned with handheld cameras in Uy et al. (2019), could further improve performance. Finally, continuing to develop new and relevant evaluation datasets if of high importance for guiding the field.

# 9 Reproducibility Statement

As demonstrated in this paper, we have been careful to conduct our work with high data standards. In continuation here of, we intend to make models, datasets, and code publicly available to foster development in the field and enable reproducibility. Similarly, we will publish all data IDs and their dataset origins used throughout the project.

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

## A  Training Dataset Details

### A.1  Pre-processing

To ensure high-quality pairing between modalities, we first filter the source datasets. We exclude text captions (Google Conceptual Captions (Sharma et al., 2018)) with poor grammar or those lacking clear references to visual or auditory concepts.

Similarly, as VidGen Tan et al. (2024) and VidRefer Yuan et al. (2025) video were originally we created for vision–text retrieval tasks, some of their videos lack a clear correspondence to the attached videos. We build a filtering pipeline to exclude such videos from our data pool.

### A.2  Pairing Modalities Through Text

We create (mod-1, text, mod-2) triples using a public (mod-1, text) dataset and a pool of mod-2 data, pairing them through text. In particular, for each unique text, we find the eight best matches in mod-2 using the models listed in Table 10, and select three of those to show the annotators for verification. The reason to start with the top eight matches is to increase diversity. We consider the (text, mod-2) candidate pairings as a bi-partite graph, with an edge between each text to the eight mod-2 candidates, we then use the greedy algorithm listed in Algorithm 1 to find a large matching between text and mod-2.

### A.3  Human Verification

We run a separate human verification project for each (mod-1, mod-2) pair, working with 18 annotators. In each project, show each annotator three (text, mod-2) candidates and ask them to label each candidate as a match, partial match, or no match. We randomize the order in which the three candidates to prevent the better candidates to appear on the same side of the screen. Even though the annotators are shown the three candidates for the same text caption all at once. Figures 4 and 5 show examples of pairs shown to the annotators in the project instructions, along with their correct labels. The label statistics for each annotation project are shown in Table 5.

**Algorithm 1** Similarity-Prioritized Greedy Matching

---

1: **Input:** Text embeddings $\mathcal{T}$, Modality 2 embeddings $\mathcal{M}$, parameters $k$, $n$, $m$
2: **Output:** Set of paired datapoints $\mathcal{P}$
3: Retrieve top $k = 8$ neighbors for each text embedding
4: Create candidate pairs $\mathcal{C} = \{(t_i, m_j, s_{ij})\}$ where $s_{ij}$ is similarity score
5: Sort $\mathcal{C}$ by similarity score in descending order
6: Initialize text_seen $\leftarrow \{\}$, modality2_seen $\leftarrow \{\}$
7: Initialize $\mathcal{P} \leftarrow \emptyset$
8: **for** each $(t, m, s) \in \mathcal{C}$ **do**
9:     **if** text_seen$[t] \geq n$ **then**
10:         **continue**
11:     **end if**
12:     **if** modality2_seen$[m] \geq m$ **then**
13:         **continue**
14:     **end if**
15:     text_seen$[t] \leftarrow$ text_seen$[t] + 1$
16:     modality2_seen$[m] \leftarrow$ modality2_seen$[m] + 1$
17:     $\mathcal{P} \leftarrow \mathcal{P} \cup \{(t, m, s)\}$
18: **end for**
19: **return** $\mathcal{P}$

---

Table 4: Counts of unique data items in each annotation project in Split 2. Leftmost column shows the datasets that contribute to each project, with the *italicized* datasets contributing the captions. & delineates the two different modalities whilst + indicates a disjoint union within the same modality.

| Dataset | Audio | Images | PCs | Captions | Video |
|---|---|---|---|---|---|
| *Valor* (Chen et al., 2023) & OpenShape | 104,832 | - | 159,810 | 104,832 | 104,832 |
| *GCC* & VGGSound + AudioSet | 91,553 | 53,478 | - | 53,478 | - |
| *Flickr* & AudioSet + VGGSound | 46,843 | 21,241 | - | 43,664 | - |
| *Audiocaps* & OpenShape | 25,434 | - | 27,994 | 24,584 | - |
| *Audiocaps* & ImageNet + GCC | 23,377 | 37,645 | - | 22,878 | - |
| *COCO* + *Flickr* & OpenShape | - | 48,487 | 107,379 | 82,989 | - |
| Total | 292,039 | 160,851 | 295,183 | 332,425 | 104,832 |

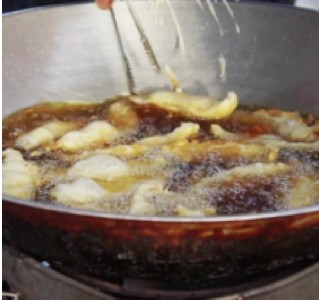
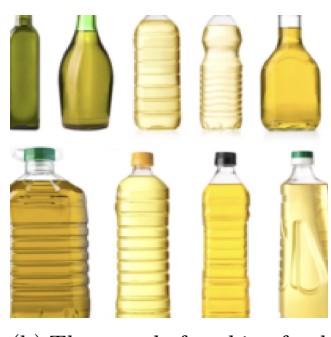
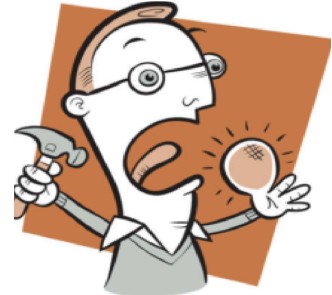

(a) The sound of cooking food in oil or another fat - Match.

(b) The sound of cooking food in oil or another fat - Partial Match.

(c) Creaky squeaking occurs as a machine runs - No Match.

Figure 4: Examples used in annotator instructions in an audio-caption-to-image pairing project

Table 5: Label statistics in each annotation project.

| Dataset | Good | Partial | No Match | Total |
|---|---|---|---|---|
| *Valor* & OpenShape | 266,686 (81.7%) | 19,244 (5.9%) | 40,464 (12.4%) | 326,394 |
| *GCC* & VGGSound + AudioSet | 68,277 (58.5%) | 22,730 (22.2%) | 22,730 (13.9%) | 163,987 |
| *Flickr* & VGGSound + AudioSet | 69,481 (48.5%) | 46,585 (32.5%) | 27,137 (27.1%) | 143,203 |
| *AudioCaps* & ImageNet + GCC | 58,536 (48.5%) | 22,246 (32.5%) | 3,129 (20.0%) | 83,911 |
| *AudioCaps* & OpenShape | 66,197 (75.5%) | – | 21,446 (24.5%) | 87,643 |
| *COCO + Flickr* & OpenShape | 232,096 (87.3%) | – | 33,769 (12.7%) | 265,865 |

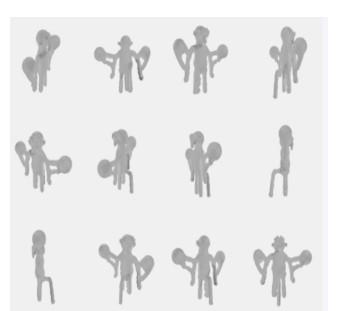

(a) A Ticktock repeats rhythmically - Match.

(b) A man clatters objects outside, making monkeys practice acrobatics with the sound - Partial Match.

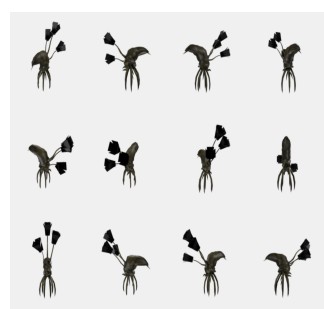

(c) A man speaking with a distant boom of a jet engine - No Match.

Figure 5: Examples used in annotator instructions in an Audio-caption-to-object pairing project

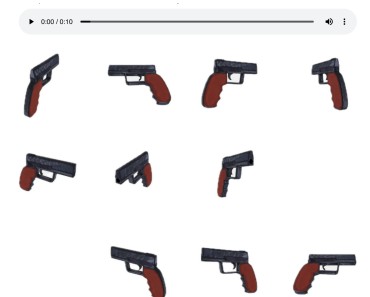

(a) 'Gunshots present in Audio Clip' - Match.

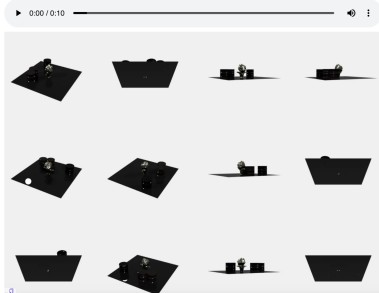

(b) 'Can hear objects being placed on table, but we would not be able to identify these specific objects from the audio file alone' - Partial Match.

Figure 6: Examples used in annotator instructions in the audio–PC pairing project

# B  ESHOT: ZERO-SHOT PC–AUDIO EVALUATION DATASET

To prevent leakage into training sets of our models as well as others, we construct our evaluation set from existing public evaluation sets. Specifically, we use the AudioSet evaluation set as our audio corpus and Objaverse LVIS as our video corpus. As these are both classification datasets, respectively containing 527 and 1,156 classes, they contain several items per class that are effectively impossible to distinguish, e.g., most sounds of a car engine can be reasonably matched with most PCs of a car. This leads us to instead develop a zero-shot PC–audio evaluation dataset.

Table 6: Text captions sourced from respective datasets.

| Dataset | Number of captions |
|---|---|
| AudioCaps | 67,335 |
| AudioSetCaps (Bai et al., 2025) (VGGSound subset) | 179,677 |
| AudiosetCaps (AudioSet subset) | 1,869,906 |
| COCO | 566,146 |
| Flickr | 147,303 |
| Google Conceptual Captions (GCC) | 524,108 |
| OpenShape (BLIP subset) | 435,507 |
| OpenShape (MSFT subset) | 335,039 |
| Valor | 1,140,010 |
| VidRefer | 360,594 |
| Vidgen | 1,004,156 |
| WavCaps | 80,779 |
| Total | 6,710,560 |

Table 7: Non-text items sourced from public datasets.

| Dataset | Audio | Images | Object | Video |
|---|---|---|---|---|
| InternVid | 1,999,975 | - | - | 1,999,960 |
| AudioSet | 1,755,876 | - | - | 1,802,084 |
| VidGen 1M | 880,733 | - | - | 896,308 |
| VidReferer | 528,493 | - | - | 518,399 |
| VGGSound | 126,465 | - | - | 167,772 |
| GCC | - | 2,229,110 | - | - |
| ImageNet | - | 1,279,867 | - | - |
| OpenShape | - | - | 827,783 | - |
| Total | 5,291,542 | 3,508,977 | 827,783 | 5,386,523 |

As with our training sets, we first automatically pair the two modalities through text captions, using SOTA retrieval models. We then run every pair through a human consensus check and admit only those pairs where three annotators label as positive matches. Figure 6 shows examples of audio–PC pairs shown to annotators in the instructions, along with their correct annotations. The consensus project yields 1,775 unique audio items, and 1,763 unique items, covering 381 LVIS classes. However, as some of the classes are indistinguishable with audio signals, we manually refine them by first correcting misclassified items, and merging indistinguishable classes. We then split broader classes into finer ones when possible—e.g., we split 'boat' into 'motor boat' and 'sail boat' classes, which are distinguishable by audio. This results in 112 classes.

Table 8: Counts of unique data (non-text) contributions of each source dataset to Split 1.

| Dataset | Audio | Images | Point Clouds | Video |
|---|---|---|---|---|
| AudioSet | 333,901 | - | - | 393,395 |
| InternVid | 174,185 | - | - | 524,514 |
| VidGen 1M | 142,441 | - | - | 368,589 |
| VidReferer | 54,632 | - | - | 144,044 |
| VGGSound | 42,327 | - | - | 35,995 |
| GCC | - | 865,303 | - | - |
| ImageNet | - | 223,779 | - | - |
| OpenShape | - | - | 209,165 | - |
| Total | 747,486 | 1,089,082 | 209,165 | 1,446,537 |

Table 9: Unique item counts of each dataset in Split 3.

| Dataset | Audio | Images | Points | Captions | Videos |
|---|---|---|---|---|---|
| Valor | 998,320 | - | - | 998,320 | 998,320 |
| AudioSet \ {Valor} | 642,574 | - | - | 642,574 | 642,574 |
| Vidgen | 408,095 | - | - | 408,095 | 408,095 |
| VidRefer | 230,883 | - | - | 230,905 | 230,883 |
| VGGSound | 120,264 | - | - | 120,264 | 120,264 |
| WavCaps | 91,482 | - | - | 91,482 | - |
| AudioCaps | 85,761 | - | - | 84,837 | - |
| Clotho | 3,838 | - | - | 19,190 | - |
| OpenShape w. Renders | - | 820,692 | 820,692 | 325,853 | - |
| Total | 2,581,217 | 820,692 | 820,692 | 3,416,359 | 2,400,136 |

Table 10: Embedding models used in the project. URLs point to model checkpoint. Index indicates when the model was used to build our search databases. Model indicates encoders used in `EBind`. A: Audio, I: Image, V: Video, P: Point Cloud, and T: Text. S1 and S2 indicate Split 1 and 2, respectively.

| Model | Index (S1) | Index (S2) | Model |
|---|---|---|---|
| PE Core-L14-336 Bolya et al. (2025) | IVT | - | IVT |
| `https://huggingface.co/facebook/PE-Core-L14-336` | | | |
| **Note:** we use `torch.amp.autocast('cuda', dtype=torch.float16)` for PE. | | | |
| EVAClip-18B Sun et al. (2024) | - | IVT | - |
| `https://huggingface.co/BAAI/EVA-CLIP-18B` | | | |
| WavCaps (HTSAT-BERT-PT) Mei et al. (2024) | AT | - | - |
| `https://drive.google.com/drive/folders/1MeTBren6LaLWiZI8_phZvHvzz4r9QeCD` | | | |
| LAION CLAP Wu et al. (2023) | - | AT | - |
| `https://huggingface.co/laion/larger_clap_general` | | | |
| ImageBind (Huge) Girdhar et al. (2023) | - | - | A |
| `https://dl.fbaipublicfiles.com/imagebind/imagebind_huge.pth` | | | |
| Uni3D (G-no-LVIS) Zhou et al. (2024) | PT | PT | P |
| `https://huggingface.co/BAAI/Uni3D/tree/main/modelzoo/uni3d-g-no-lvis` | | | |

## C  EVALUATION DETAILS

Here we list the non-trivial processing steps we followed when evaluating our models.

**Retrieval:**  Clotho, COCO, and Flickr30K, all have multiple text captions for each data item of the other modality. In calculating text recall@k from the other modality, we considered retrieval a success if any of the captions were in the top-k. For the opposite retrieval direction, we considered each caption to be a separate item and averaged their recall results as usual. To the best of our understanding, this is consistent with how all other models we benchmark against have been evaluated.

**Zero-shot Classification:**  We found , we found prompt templating to help on ImageNet and ModelNet40. That is, instead of using a single prompt to calculate the embedding for each class, we calculated the average embeddings over several prompt templates and took the mean embedding as the class representative. In particular, we used Perception Encoder's prompt templates (Bolya et al., 2025, Appendix B.1.2) for ImageNet and PointBind's templates[3] for ModelNet40.

## D  USAGE OF LARGE LANGUAGE MODELS

As in most modern workflows, we use large language models (LLMs) to conduct our work. While tap-completions have been enabled in our text editors, we have not included work solely done by LLMs on the behalf of Humans. In other words, LLMs have been used for writing template code and expand some bulleted lists into first drafts of paper sections. However, no sentence is without human involvement and no citations were added by LLMs. In continuation here of, every citation has been verified by humans. We are confident that we have operated well within what is considered appropriate for the conference.

---

[3]`https://github.com/ZiyuGuo99/Point-Bind_Point-LLM/blob/main/data/templates.json`

Table 11: Benchmarks used to evaluate `EBind`. Zero-shot means zero-shot classification and Retrieval means cross-modal retrieval between two modalities.

| Task | Modality | Benchmark | Items | Classes |
|------|----------|-----------|-------|---------|
| Zero-shot | Audio | AudioSet Gemmeke et al. (2017) | 17,141 | 527 |
| | | ESC-50 Piczak (2015) | 2,000 | 50 |
| | Image | ImageNet1K Deng et al. (2009) | 50,000 | 1000 |
| | | Objaverse-LVIS Deitke et al. (2023) | 46,205 | 1156 |
| | Points | ScanObjNN Uy et al. (2019) | 2,890 | 15 |
| | | ModelNet40 Wu et al. (2015) | 2,467 | 40 |
| | Audio-Points | EShot (ours) | 3,538 | 112 |
| Retrieval | Audio-Text | AudioCaps Kim et al. (2019) | 957 | - |
| | | Clotho Drossos et al. (2020) | 27,905 | - |
| | Audio-Image | VGG-SS Chen et al. (2021) | 5,116 | - |
| | | FlickrNet Senocak et al. (2018) | 5,000 | - |
| | Audio-Text | COCO Lin et al. (2014) | 5,000 | - |
| | | Flickr30K Young et al. (2014) | 1,000 | - |
| | Points-Image | Objaverse-LVIS Deitke et al. (2023) | 46,205 | - |