# OpenReview forum: "EBind: a Practical Approach to Space Binding"
_ICLR.cc/2026/Conference — ICLR 2026 Conference Withdrawn Submission_

### Official Review · Reviewer_HNsA · 2025-10-27

**Soundness:** 2
**Presentation:** 2
**Contribution:** 2
**Rating:** 2
**Confidence:** 5

**Summary:**

This paper strives to simplify space binding by focusing on single encoder per modality and high quality dataset. EBind is introduced as a data-centric, parameter-efficient method to bind the embedding spaces of multiple contrastive models.

**Strengths:**

1. The proposed EBind is somehow parameter efficient.

2. The three complementary data sources is a good contribution which inlcude 6.7 M fully-annotated multimodal quintuples, 1 M diverse semi-automated triples annotated by humans, and 3.4 M pre-existing captioned data items.

**Weaknesses:**

1. This paper is more like a "Dataset & Benchmark Track" submission, the reviewer believe the technical contribution is limited.

2. In the experiment section, only several Bind model are included, more should be compared and discussed:

[A] Girdhar R, El-Nouby A, Liu Z, et al. Imagebind: One embedding space to bind them all[C]//Proceedings of the IEEE/CVF conference on computer vision and pattern recognition. 2023: 15180-15190.

[B]  Zhu B, Lin B, Ning M, et al. LanguageBind: Extending Video-Language Pretraining to N-modality by Language-based Semantic Alignment[C]//The Twelfth International Conference on Learning Representations.

[C] Lyu Y, Zheng X, Zhou J, et al. Unibind: Llm-augmented unified and balanced representation space to bind them all[C]//Proceedings of the IEEE/CVF Conference on Computer Vision and Pattern Recognition. 2024: 26752-26762.

[D] Wang Z, Zhang Z, Cheng X, et al. FreeBind: Free Lunch in Unified Multimodal Space via Knowledge Fusion[C]//Forty-first International Conference on Machine Learning.

**Questions:**

1. On the Paper’s Contribution Scope and Novelty

Q1: Could the authors clarify how this work extends beyond being primarily a Dataset & Benchmark Track contribution?

Q2: What specific technical or methodological contributions distinguish this paper from a pure data/benchmark release?

Q3: How does the proposed benchmark advance multimodal learning compared with existing datasets (e.g., ImageBind or LanguageBind)?

Q4: Are there any underlying design principles or architectural choices that introduce new insights into model evaluation, data organization, or modality alignment?

Q5: How should the paper be positioned within the community — as an infrastructure enabler or as a technical contribution introducing new methods?

2. On Experimental Evaluation and Comparative Analysis

Q6: The current experimental section includes only a few baseline models. Could the authors expand the comparison to include more Bind-family methods, such as:

[A] Girdhar et al., ImageBind: One embedding space to bind them all, CVPR 2023

[B] Zhu et al., LanguageBind: Extending Video-Language Pretraining to N-Modality by Language-based Semantic Alignment, ICLR 2024

[C] Lyu et al., UniBind: LLM-Augmented Unified and Balanced Representation Space to Bind Them All, CVPR 2024

[D] Wang et al., FreeBind: Free Lunch in Unified Multimodal Space via Knowledge Fusion, ICML 2024

Q7: Were these additional models excluded due to resource limitations or compatibility concerns?

Q8: Are all compared methods trained or evaluated under a consistent setup (e.g., modality combinations, training data, and loss definitions)?

Q9: Can the authors provide further ablation or breakdown results per modality (e.g., RGB, depth, audio) to strengthen fairness and interpretability?

Q10: Would the inclusion of qualitative analyses—such as failure cases, visualization of retrieval alignments, or representation projections—help illustrate model behavior differences?

---

### Official Review · Reviewer_f595 · 2025-10-30

**Soundness:** 3
**Presentation:** 3
**Contribution:** 2
**Rating:** 6
**Confidence:** 4

**Summary:**

EBind proposes a data-driven lightweight multimodal embedding space fusion method, mapping five modalities—images, text, audio, video, and 3D point clouds (PC)—into a unified embedding space. The paper designs a minimalist architecture and trains it with data in three stages. In addition, the paper introduces the first audio-PC zero-shot classification benchmark. EBind achieves high performance in multiple cross-modal tasks with a relatively low number of model parameters, demonstrating its efficiency.

**Strengths:**

The paper is innovative, designing a new model architecture and filling the gap in audio-point cloud data evaluation. The structure of the paper is clear, with intuitive figures and a detailed appendix, including annotation examples. The experimental quality of the paper is good, but some ablation experiments are missing. The significance of the paper is notable.

**Weaknesses:**

1. Since the visual-text model is frozen, the improvement in visual capabilities relative to other models entirely depends on the selection of a new base model. In fact, to make a fair comparison, it would be best to add an ablation experiment using CLIP-L from ImageBind as the visual-text model. This would not diminish the superiority of EBind in current setting.

2. As mentioned above, the comparison with Ex-MCR is unfair and incomplete. Ex-MCR provides a version based on CLIP-L and combines it with ImageBind. It is hoped that the authors will include this comparison.

3. The innovation of the EBind method mainly comes from the combination of various data collection methods. Its model architecture, training method, and computational overhead are similar to those of projector-based models such as Ex-MCR. EBind has an advantage in model parameters compared to works like OmniBind, but it seems to overlook the motivation behind the low-cost data construction methods of Ex-MCR and OmniBind. In fact, as mentioned in C-MCR, the biggest drawback of relying on existing data pairs is scalability and generalizability. Once a modality lacks diverse native paired data, the alignment quality of the modality will decline significantly (the experimental results of S1, S2, and S3 in the paper also indicate this). In addition, EBind's split2 requires a large amount of human annotation, which may reduce the scalability of EBind's data pipeline.

Ex-MCR: https://proceedings.neurips.cc/paper_files/paper/2024/file/a71df365f872a39e58475f1fa7950879-Paper-Conference.pdf

**Questions:**

1. How are the experimental results of EBind when using the base CLIP-L? Can effective improvements be achieved?

2. How are the experimental results of other projector-based methods when using EBind's data?

---

### Official Review · Reviewer_H74Q · 2025-10-30

**Soundness:** 3
**Presentation:** 2
**Contribution:** 2
**Rating:** 2
**Confidence:** 4

**Summary:**

- The paper introduces EBind, a parameter-efficient method designed to bind the embedding spaces of multiple contrastive models across five modalities: image, text, video, audio, and 3D point clouds.

- This model achieves descent performance relative to its size, often competitive to larger models, and democratizes training by achieving results on a single GPU in hours.

- The core methodology relies on leveraging a simple architecture with frozen encoders and small projectors, coupled with a systematic three-tier data curation strategy that includes 1M human-annotated samples with partial and negative match labels.

- The authors address evaluation limitations by introducing EShot, a zero-shot classification benchmark specifically for audio and PC pairs.

**Strengths:**

- The proposed model achieves its parameter efficiency by utilizing a simple architecture based on frozen, pre-trained encoders for all five modalities. Leading to light weight architecture for simplicity and achieved descent results to be obtained on a single GPU within hours.

- The work introduces a consensus-annotated zero-shot classification benchmark, EShot, for the audio-Point Cloud modality pair, alongside the commitment to open-sourcing the code, model weights, and curated datasets for reproducibility.

**Weaknesses:**

1. The model demonstrates low performance in cross-modal Audio-Text retrieval tasks (such as AudioCaps and Clotho) compared to larger competitor models, a gap the authors hypothesize is due to their choice of using the ImageBind audio encoder, which was initially optimized against images rather than text. In that case, why did the binding strategy not alleviate such an issue? Why was this specific, weakly-performing encoder chosen over potentially stronger, text-aligned audio encoders like CLAP or WavCaps for the final model architecture?

2. The pursuit of flexibility and computational efficiency leads the model to take less than full advantage of its automatically paired training data (Split 1), as the architecture requires training each projector separately, meaning that only one of the two projected modalities enters training for each projector, despite the availability of 6.7M 5-tuple data. What will be a good way to scale up to more modality pairs?

3. The model exhibits weaker zero-shot classification capability for audio and a lower score on the ScanObjectNN Point Cloud benchmark compared to its base encoder Uni3D. The author stated that there is no explanation for why this happens, resulting in a lack of insight in this paper.

4. The final training stage on Split 3, which includes naturally paired data but lacks explicit Audio-PC pairings, hurts the R@1 performance of the EShot benchmark for Audio-Points, indicating a problem of "forgetting" previous joint modality learning when new data without those specific pairs is introduced. This raises a significant question about the generalizability of this method when encountering new modalities and tasks.

**Questions:**

1. Please answer the questions in the weakness section.

2. Given the model's efficiency relies on handling videos by uniformly sampling 8 frames and averaging their embeddings, did the authors investigate if a dedicated, small trainable video projector or a temporal pooling mechanism beyond simple averaging could enhance performance, particularly since video benchmarks were omitted from the main evaluation?

3. The training process uses three distinct data splits, with two epochs allocated to each split sequentially. Were ablation studies performed to determine if altering the number of epochs per stage or introducing loss weighting across stages could mitigate the R@1 performance drop on EShot observed when transitioning to the naturally paired data in Split 3?

---

### Official Review · Reviewer_U31C · 2025-11-02

**Soundness:** 2
**Presentation:** 2
**Contribution:** 3
**Rating:** 4
**Confidence:** 5

**Summary:**

Compared with ImageBind and similar methods, EBind achieves strong cross-modal alignment within **<6 hours on a single GPU** by prioritizing **high-quality training data**. Its effectiveness is validated across multimodal retrieval and classification tasks.

**Strengths:**

1. Rather than merely scaling data and model size, the work explores a **parameter-efficient strategy** to achieve superior performance, which represents a more sustainable research direction.
2. EBind can be trained rapidly while attaining performance competitive with OmniBind, demonstrating **high efficiency without sacrificing accuracy**.
3. The paper also provides **practical applications** that showcase real-world utility.

**Weaknesses:**

1. The single-GPU and fast-training advantages are largely attributed to very few trainable parameters and extensive data preprocessing. Could PointBind or OmniBind achieve similar results under the same constraints?
2. The training uses **nearly 2M paired samples**. Is the performance primarily due to data scale rather than data quality? How does the dataset size compare to other methods?
3. Can the trained model be integrated with current generative models or LMMs? If so, please provide examples and quantitative improvements.

**Questions:**

1. **On the claimed efficiency (single GPU & fast training):**
   The paper attributes its efficiency primarily to freezing most encoders, using only lightweight projection heads, and relying on extensive offline embedding preprocessing. While this design is appealing, it raises the question of whether the observed advantages are **unique to EBind’s architecture**. In principle, existing models such as PointBind or OmniBind could potentially adopt similar strategies—e.g., freezing encoders and training only projection layers—to achieve comparable efficiency.

   * Have the authors tested or benchmarked a “frozen-encoder + projector-only” variant of these baselines?
   * Without such comparisons, it is difficult to conclude that the efficiency gain arises from the proposed method, rather than simply from reducing the number of learnable parameters.

2. **Data scale vs. data quality:**
   The method uses approximately **2M paired training samples**, combining retrieved, human-annotated, and curated datasets. Although the paper emphasizes “data quality” as the key driver of performance, such a large-scale dataset could by itself substantially contribute to the strong results.

   * Is there an **ablation study** demonstrating that high-quality human-verified samples outperform an equal-volume random or synthetic dataset?
   * How does EBind’s total data volume compare with OmniBind, PointBind, and other prior works?
   * Without clearer evidence, it remains ambiguous whether the improvements stem primarily from data quality or simply from **big data**.

3. **Compatibility with generative models / LMMs:**
   Since modern multimodal research increasingly focuses on **generative capabilities** and integration with large multimodal models (LMMs), it would be valuable to understand whether EBind can interface with such systems.

   * Can the learned cross-modal embedding space be leveraged by current generative models (e.g., for audio- or point-cloud-conditioned generation)?
   * Are there examples demonstrating improved downstream performance when EBind embeddings are plugged into an LMM or diffusion-based model?
   * If so, please provide **quantitative results or case studies**. If not, a discussion on feasibility and limitations would be helpful.

---

### Author Response · Authors · 2025-11-18
**[withdrawing] Thank you to all reviewers**

We would like to extend our gratitude to all four reviewers for their thoroughness, constructive feedback, valid concerns, and references to additional related work.

We have taken notes of the feedback spanning evaluation considerations, disentangling data scale and quality, performance issues on certain tasks, indications of "catastrophic forgetting," and the limitations on technical novelty. We genuinely believe that our work demonstrates how technical novelty is not strictly necessary to obtain good model performance and thus provides valuable information to the field. While we demonstrate how both data scale and quality can provide significant performance gains, we recognize that perhaps this work is better positioned for a dataset/benchmark track.
a
We intend to revise and expand our work based on the feedback to publish it elsewhere.

Once again, a big thank you to everyone involved.

~ The authors

---

### Note · Authors · 2025-11-18

**Comment:**

We would like to extend our gratitude to all four reviewers for their thoroughness, constructive feedback, valid concerns, and references to additional related work.

We have taken notes of the feedback spanning evaluation considerations, disentangling data scale and quality, performance issues on certain tasks, indications of "catastrophic forgetting," and the limitations on technical novelty. We genuinely believe that our work demonstrates how technical novelty is not strictly necessary to obtain good model performance and thus provides valuable information to the field. While we demonstrate how both data scale and quality can provide significant performance gains, we recognize that perhaps this work is better positioned for a dataset/benchmark track. a We intend to revise and expand our work based on the feedback to publish it elsewhere.

Once again, a big thank you to everyone involved.

~ The authors

**Withdrawal Confirmation:**

I have read and agree with the venue's withdrawal policy on behalf of myself and my co-authors.